# Phenotyping the Histopathological Subtypes of Non-Small-Cell Lung Carcinoma: How Beneficial Is Radiomics?

**DOI:** 10.3390/diagnostics13061167

**Published:** 2023-03-18

**Authors:** Giovanni Pasini, Alessandro Stefano, Giorgio Russo, Albert Comelli, Franco Marinozzi, Fabiano Bini

**Affiliations:** 1Department of Mechanical and Aerospace Engineering, Sapienza University of Rome, Eudossiana 18, 00184 Rome, Italy; 2Institute of Molecular Bioimaging and Physiology, National Research Council (IBFM-CNR), Contrada, Pietrapollastra-Pisciotto, 90015 Cefalù, Italy; 3Ri.MED Foundation, Via Bandiera 11, 90133 Palermo, Italy

**Keywords:** radiomics, CT, non-small-cell lung carcinoma, NSCLC, phenotyping, multicenter, harmonization, machine learning

## Abstract

The aim of this study was to investigate the usefulness of radiomics in the absence of well-defined standard guidelines. Specifically, we extracted radiomics features from multicenter computed tomography (CT) images to differentiate between the four histopathological subtypes of non-small-cell lung carcinoma (NSCLC). In addition, the results that varied with the radiomics model were compared. We investigated the presence of the batch effects and the impact of feature harmonization on the models’ performance. Moreover, the question on how the training dataset composition influenced the selected feature subsets and, consequently, the model’s performance was also investigated. Therefore, through combining data from the two publicly available datasets, this study involves a total of 152 squamous cell carcinoma (SCC), 106 large cell carcinoma (LCC), 150 adenocarcinoma (ADC), and 58 no other specified (NOS). Through the matRadiomics tool, which is an example of Image Biomarker Standardization Initiative (IBSI) compliant software, 1781 radiomics features were extracted from each of the malignant lesions that were identified in CT images. After batch analysis and feature harmonization, which were based on the ComBat tool and were integrated in matRadiomics, the datasets (the harmonized and the non-harmonized) were given as an input to a machine learning modeling pipeline. The following steps were articulated: (i) training-set/test-set splitting (80/20); (ii) a Kruskal–Wallis analysis and LASSO linear regression for the feature selection; (iii) model training; (iv) a model validation and hyperparameter optimization; and (v) model testing. Model optimization consisted of a 5-fold cross-validated Bayesian optimization, repeated ten times (inner loop). The whole pipeline was repeated 10 times (outer loop) with six different machine learning classification algorithms. Moreover, the stability of the feature selection was evaluated. Results showed that the batch effects were present even if the voxels were resampled to an isotropic form and whether feature harmonization correctly removed them, even though the models’ performances decreased. Moreover, the results showed that a low accuracy (61.41%) was reached when differentiating between the four subtypes, even though a high average area under curve (AUC) was reached (0.831). Further, a NOS subtype was classified as almost completely correct (true positive rate ~90%). The accuracy increased (77.25%) when only the SCC and ADC subtypes were considered, as well as when a high AUC (0.821) was obtained—although harmonization decreased the accuracy to 58%. Moreover, the features that contributed the most to models’ performance were those extracted from wavelet decomposed and Laplacian of Gaussian (LoG) filtered images and they belonged to the texture feature class.. In conclusion, we showed that our multicenter data were affected by batch effects, that they could significantly alter the models’ performance, and that feature harmonization correctly removed them. Although wavelet features seemed to be the most informative features, an absolute subset could not be identified since it changed depending on the training/testing splitting. Moreover, performance was influenced by the chosen dataset and by the machine learning methods, which could reach a high accuracy in binary classification tasks, but could underperform in multiclass problems. It is, therefore, essential that the scientific community propose a more systematic radiomics approach, focusing on multicenter studies, with clear and solid guidelines to facilitate the translation of radiomics to clinical practice.

## 1. Introduction

Lung cancer is the second most common type of cancer, both in men and in women. Moreover, despite its incidence rate being less than prostate cancer and breast cancer, its mortality rate is higher. Indeed, lung cancer is the leading cause of cancer deaths in 2022 [1,2]. However, the percentage of people still living 3 years after diagnosis is increasing, especially thanks to advances in surgical techniques, early detection, and targeted therapies. However, this is largely confined to non-small-cell lung cancer (NSCLC) [1].

NSCLC is the type of lung cancer that occurs most often (85%) between the two main forms of lung cancer—which are NSCLC and small-cell lung cancer (SCLC). Moreover, the World Health Organization (WHO) has further classified NSCLC into three main groups: adenocarcinoma (ADC, ~40%), squamous cell carcinoma (SCC, 25–30%), and large cell carcinoma (LCC, 5–10%). Furthermore, difficulties in the NSCLC diagnosis have led to the creation of a fourth class, named “not otherwise specified” (NOS), which includes NSCLCs that do not have the characteristics of the three main subtypes [3,4].

Typically, NSCLC is diagnosed in cases of advanced-stage disease, which is when patients that show symptoms—such as coughing, hemoptysis chest pain, and dyspnea—undergo medical imaging that is followed by biopsy. Furthermore, the efficacy of treatment strongly depends on the cancer stage, and it is mostly based on surgery, chemotherapy, immunotherapy, and radiotherapy [3,5].

Recently, precision medicine has emerged as an innovative approach to refine the treatment of NSCLC, according to histological and molecular subtypes, thus improving disease outcome [6]. However, the early detection, differential diagnoses of NSCLC subtypes, and the cancer staging remain the main challenges in terms of allowing treatment personalization. To address this issue, radiomics could be a useful tool to support the clinical decision process [7].

Radiomics is an emerging research field that involves the use of artificial intelligence to analyze medical images, either through using classical machine learning pipelines or through advanced deep learning methods. Its aim is to extract quantitative metrics that can be used to build predictive models that are able to respond to a specific clinical question. Therefore, radiomics can be used to identify the most predictive biomarkers in a disease, to perform differential diagnoses between cancer subtypes and lung diseases, and to predict overall survival and responses to therapy [8,9,10]. Radiomics is used to obtain information that is not even visible to the eyes of expert clinicians and radiologists, such as the shape, the distribution of gray levels, and the texture of a lesion, thus increasing the amount of data available. Its main advantage over biopsy is that it is non-invasive, less time consuming, and can be integrated in automated pipelines [11].

However, despite the Imaging Biomarker Standardization Initiative (IBSI) [12] being provided, the guidelines on how radiomics features should be computed, the lack of standardization in the parameters used during the feature extraction process (as well as in the algorithms that are used to perform feature selection), and machine learning are all still major challenges and represent the limits of radiomics itself [13].

Moreover, when medical images come from different centers, dissimilarities in the protocols and in the technical specifications of the imaging manufacturers can lead to the generation of batch effects. This can have a negative impact on radiomics analysis. To address this issue, batch analysis and feature harmonization are needed [14,15].

In oncology, several studies demonstrated the predictive and prognostic power of radiomics, such as those focused on the detection and localization of prostate cancer, the Gleason Score (GS), and the recurrence prediction [16,17], while others were focused on the differential diagnoses between lung cancer types and the prediction of lung nodules’ malignancy [18,19]. Radiomics was also applied to breast cancer detection and several studies demonstrated that its integration with mammography and magnetic resonance imaging (MRI) improved diagnostic accuracy [20]. Other applications are those that involve radiomics for the evaluation of both gastrointestinal tumors [21] and brain tumors [22]. In neuroscience, radiomics is applied for the early detection of neurodegenerative diseases, such as Alzheimer’s [23] and Parkinson’s [24] disease. In addition, many deep learning methods have been proposed [25], such as those that were developed for automated multiple sclerosis detection [26]. It has also been applied to predict the expanded disability status scale (EDSS) [27] of patients who were affected by multiple sclerosis. Meanwhile, certain studies also focused on the prediction of epilepsy in patients affected by frontal glioma [28].

Regarding NSCLC, radiomics studies have focused on predicting mediastinal lymph node metastasis [29]. More specifically, they were mostly focused on only differentiating between the two NSCLC subtypes, such as SCC and ADC [30,31,32,33,34], rather than focusing on multi-subtype classification [35,36], which is crucial since treatment strategies strongly depend on the NSCLC subtype. Moreover, the differentiation between lung adenocarcinoma and lung squamous cell carcinoma subtypes has also been investigated in the field of genomics, and it was through a novel explainable AI(XAI)-based deep learning framework [37] that promising results were achieved.

### 1.1. Related Works

In this section, we divided the existing studies related to NSCLC classification problems into two groups: those whose aim was a multiclass classification problem (i.e., four classes: NOS, ADC, LCC, and SCC) and those whose aim was a binary classification problem (i.e., two classes: SCC and ADC). We summarize their main characteristics in Table 1 and Table 2. Meanwhile, the comparison between machine learning methods and results will be illustrated in Section 4.1. Moreover, we limit the comparison to studies that only used classical machine learning models, as we did not use deep learning in our study. Table 1 shows that two studies that evaluated binary classification were multicenter studies [33,34], but none of them investigated the presence of batch effects, and neither performed feature harmonization. Moreover, certain details regarding image pre-processing were lacking in all the studies—except in Haga et al. [31]—which was particularly related to the software that was used for the feature extraction and for the extracted features. For example, discretization was needed for the extraction of texture features [12], but discretization parameters were lacking in [30,32,33,34]. The only study that correctly reported the discretization parameters (bin count 225, bin size 25 HU) was [31]. Moreover, in certain studies [32,33,34], whose filtering and/or wavelet decomposition were applied, the parameters regarding filters and the method for wavelet decomposition were lacking. Indeed, only two studies [30,31] reported the method that was used for wavelet decomposition (namely, coiflet). Furthermore, the isotropic voxel dimension was correctly reported in all the studies in which images were resampled, but only in Song et al. [34] was the interpolator (namely, bilinear interpolation) specified. Neither of the studies [35,36] in Table 2 specified all of the parameters that were used for the pre-processing. Indeed, in Khodabakhshi et al. [36], the isotropic voxel (1 mm^3^), discretization parameters (64 bins), and sigma (0.5–5; 0.5 step) that were used for the LoG filter were specified, but the methods used for the wavelet decomposition and the spatial resampling were missing. Meanwhile, in Liu et al. [35], the wavelet decomposition method (namely, coiflet) was specified, but no information about discretization or feature extractor were given. Moreover, both studies were not multicenter studies, because only one dataset was used in [36], and two datasets, but coming from the same center (i.e., the MAASTRO clinic), were used in [35]. Finally, none of the studies that were based on PyRadiomics as the extractor specified if the remaining parameters were set to PyRadiomics’ default values.

### 1.2. Research Motivation and Contribution

Even if IBSI [12] provides guidelines on how to compute radiomics features, they are still strongly influenced by the software that is used and the pre-processing parameters. This aspect, together with a non-exhaustive report of the extraction parameters, makes radiomics studies less robust and more difficult to reproduce. Moreover, few studies have focused on the multiclass classification of NSCLC subtypes, while many of them only focused on differentiating between the two subtypes. Furthermore, only a few studies (e.g., [34]), investigated the potentiality of models that were based on multicenter data, which we strongly believe is the next frontier for the creation of more robust and generalizable models. To the best of our knowledge, this is the first study which investigated the presence of batch effects and the impact of feature harmonization on multicenter radiomics CT-based models for the phenotyping of four different NSCLC subtypes.

Following this last line of research, this study
Investigates the possibility of building a multiclass multicenter radiomics-based machine learning model, which is capable of differentiating between the NSCLC subtypes with the aim of improving treatment personalization;Evaluates the presence of batch effects in a multicenter study with the aim to assess the impact of feature harmonization on machine learning models;Evaluates the stability of the feature selection procedure by iterating the machine learning modeling pipeline 10 times with the aim of reducing the result variability;Shows that selected feature subsets are influenced by training/testing set splitting;Provides a scientific and critical analysis of the advantages and disadvantages of radiomics in the absence of well-defined standard guidelines with the aim of promoting the need for reproducible and repeatable radiomics studies.

The article is organized as follows: the used datasets, image segmentation, image pre-processing, feature extraction, batch analysis, feature harmonization, and the machine learning modeling pipeline are all described in the Section 2. The feature stability, feature harmonization, and model performance results are described in the Section 3. The Appendix A are provided in the Appendix A. The Section 4 and Section 6 provide explanations for the obtained results in comparison with previous studies, and with the current issues of radiomics. A recap of the relevant feature extraction parameters is given in Appendix B, while the acronyms are reported in Appendix C.

## 2. Materials and Methods

### 2.1. Data Preparation and Image Segmentation

The medical images used in this study were obtained by combining the data from two publicly available datasets, namely the NSCLC-Radiomics [38] dataset and the NSCLC-Radiogenomics dataset [39]. The data preparation workflow is illustrated in Figure 1.

### 2.2. NSCLC-Radiomics Dataset

The NSCLC-Radiomics dataset contains CT images from 422 non-small-cell lung cancer patients and their associated segmentations, both in the digital imaging and communication in medicine (DICOM) format. Meanwhile, the histopathological information is provided in a .xls file. The segmentations were manually performed by experts [38]. The original dataset contains 152 patients who were diagnosed with SCC, 114 patients that were diagnosed with LCC, 51 patients who were diagnosed with ADC, 63 patients specified as NOS, and 42 patients who were specified as NA (because the diagnosis was not available). The NA cases were excluded from the original dataset. Using the matRadiomics software [18], a visual inspection of all the cases and their associated segmentation was carried out. Due to the quality of images and segmentations, the total number of patients was further reduced to 324, composed of the 123 SCC cases, 106 LCC cases, 40 ADC cases, and the 55 NOS cases.

### 2.3. NSCLC-Radiogenomics

The NSCLC-Radiogenomics dataset contains computed tomography images from 144 non-small-cell lung cancer patients and their associated segmentations, both in the DICOM format. Meanwhile, the histopathological information is provided in a .csv file. The segmentations were performed through a semiautomatic algorithm and were manually refined by an expert [39]. The original dataset contains 29 patients who were diagnosed with SCC, 112 patients diagnosed with ADC, and 3 patients specified as NOS. Two patients that belonged to the ADC class were excluded from the original dataset after visual inspection through matRadiomics. Therefore, we only considered a total of 142 patients.

#### 2.3.1. CT Images’ Pixel Spacing, Slice Thickness, Matrix Dimension, and Manufacturers

Since the images were acquired in different institutions—i.e., the MAASTRO clinic [46] (NSCLC-Radiomics dataset), the Stanford University Medical Center, and the Palo Alto Veterans Affairs Healthcare System (VA) (NSCLC-Radiogenomics dataset)—different scanning devices were used. Therefore, we inspected the DICOM attributes related to each CT scan through the matRadiomics software to investigate if differences in the pixel spacing, slice thickness, and matrix dimensions were present [47]. The data are shown in Table 3.

As shown in Table 3, the differences in pixel spacing and slice thickness were individuated. Moreover, the 247 CT scans with [0.9765625, 0.9765625] pixel spacing were acquired using Siemens devices, while the 77 CT scans with [0.9770, 0.9770] pixel spacing were acquired using CMS devices, as reported in the manufacturer DICOM attribute (tag 0008, 0070); furthermore, they all belonged to the NSCLC-Radiomics dataset. Since in the NSCLC-Radiogenomics dataset voxel spacing and slice thickness values varied greatly for each patient, the division in the groups was not carried out and the values were reported within a range. In this case, the images of the 142 patients were acquired using Siemens, GE Medical Systems, and Philips devices. Therefore, four different device manufacturers were individuated: Siemens, CMS, GE Medical Systems, and Philips.

#### 2.3.2. Image Pre-Processing and Feature Extraction

Using matRadiomics, 1781 radiomics features were extracted from each of the patients included in this study. All the quantitative metrics can be grouped into three major classes: (i) shape/morphological features, (ii) first order statistical features, and (iii) texture features, which are the gray level co-occurrence matrix (GLCM), gray level run length matrix (GLRLM), gray level size zone matrix (GLSZM), neighboring gray tone difference matrix (NGTDM), and the gray level dependence matrix (GLDM)). Since matRadiomics uses the PyRadiomics [19] package to perform feature extraction, the image pre-processing parameters were set according to a previous study [36]. The parameters used for image pre-processing are reported as follows: the voxels were resampled to an isotropic voxel (1 × 1 × 1 mm^3^), using the *sitkLinear* interpolator, while the gray levels were discretized into 64 bins using the *bin count* option. Prior to feature extraction, the LoG was used to filter the original images with different *sigma* values (0.5, 1, 1.5, 2, 2.5, 3, 3.5, 4, 4.5, and 5), and the 8 wavelet Haar transform was used to decompose (HHH (high-high-high), HHL (high-high-low), HLH (high-low-high), LHH (low-high-high), LLL (low-low-low), LHL (low-high-low), HLL (high-low-low), and LLH (low-low-high)) the original images. All the other parameters were left to PyRadiomics’ default (https://pyradiomics.readthedocs.io/en/latest/customization.html#feature-extractor-level, accessed on 1 January 2023). A recap of all the feature extraction parameters is provided in Table A1 in Appendix B. The application of the LoG filter and the 8-wavelet decomposition made it possible to obtain the first order statistical features, as well as the texture features that were computed both on the LoG-filtered images and on the decomposed images, not only on the original images. The feature extraction workflow is schematically illustrated in Figure 2.

#### 2.3.3. Dataset Preparation for Batch Analysis, Feature Harmonization, and for the Machine Learning Modeling Pipeline

After feature extraction, the combined dataset, containing the 1781 features that were extracted for each of the 466 patients, was further subdivided into three datasets. The first one, namely the “4-classes dataset (4-c)”, contains all the classes (SCC, LCC, ADC, and NOS) of the combined dataset, and essentially coincides with it. The second one, namely “3-classes dataset (3-c)”, contains only three classes (SCC, LCC, and ADC), while the third one, namely “2-classes dataset (2-c)”, contains only the most numerous classes (SCC and LCC). The final dataset subdivision is shown in Figure 3.

#### 2.3.4. Batch Analysis and Feature Harmonization

Since CT images were obtained in three different centers (MAASTRO, Stanford, and VA) while using four different scanners, we investigated the presence of the batch effects. This analysis was preliminarily conducted on all the datasets (the 4-classes dataset, 3-classes dataset, and the 2-classes dataset) and results were used to know which classes to use in order to build the batch vector, which was further used to perform feature harmonization across all the datasets. Therefore, principal component analysis (PCA) was performed to project the data in a space of reduced dimensions. We further performed the visual inspection of the 2D plot, consisting of only the first and second principal components (PC1, PC2) in order to investigate the presence of clusters. Moreover, we used the t-distributed stochastic neighbor embedding (tSNE) to check for the batch effects using four different distance methods: Euclidean, cityblock, Minkowski, and Chebychev. Finally, the Kruskal–Wallis test was performed to assess if a statistically significant difference was present between the clusters. To assess which groups were significantly different, the Kruskal–Wallis test was followed by the post-hoc Tukey-HSD test. Therefore, to remove the batch effects, we performed feature harmonization through the ComBat package (https://github.com/Jfortin1/ComBatHarmonization, accessed on 1 January 2023), which was integrated in matRadiomics and aim of which was to standardize the mean and variance across the batches. Finally, we verified if the batches were correctly removed and whether they performed feature harmonization on all of the datasets. Therefore, three more datasets were generated, the 4-classes harmonized dataset (4-c-h), the 3-classes harmonized dataset (3-c-h), and the 2-classes harmonized dataset (2-c-h). The Figure 4 pipeline shows the batch analysis and feature harmonization workflows.

#### 2.3.5. Machine Learning Modeling Pipeline

All the datasets, both harmonized and non-harmonized, were given as the input to the machine learning modeling pipeline. The adopted pipeline consists in repeating the following scheme 10 times (i.e., the outer loop technique): (i) training-set/testing-set stratified splitting (80/20 ratio), (ii) a Kruskal–Wallis analysis followed by the well-known least absolute shrinkage and selection operator (LASSO) [48,49,50] for feature selection, (iii) model training, (iv) model validation and hyperparameter optimization, and (v) model testing.

The machine learning modeling pipeline is illustrated in Figure 5.

### 2.4. Feature Selection and Feature Stability

Since all the datasets, both non-harmonized and harmonized, are high-dimensional, with a number of features (1781) that is much greater than the number of cases (466, 408 and 302), feature reduction and selection are needed. This procedure was carried out only on the training sets. Moreover, given the results on the training sets, the testing sets were adjusted leaving the selected features and discarding the rest. A Kruskal–Wallis analysis was only used to retain the features with a *p*-value that was less than a specified threshold. Therefore, three increasing thresholds were set (*p*-value < 0.005, *p*-value < 0.01, and *p*-value < 0.05) and the following scheme was adopted to switch between the thresholds: (i) If no feature meets the first threshold (*p* < 0.005), then the *p*-value increases to the second threshold; (ii) if no feature meets the second threshold, then the *p*-value increases to the third threshold; and (iii) if none of the features meets the third threshold, then all the features are given as an input to the LASSO.

LASSO, a well-known feature selection algorithm in radiomics, was used to select the most important features. In addition, the optimal value of lambda was obtained through a 10-fold cross validation procedure. Finally, we obtained ten subsets of the selected features, and for each selected feature we computed its frequency. A 100% frequency means that the selected feature appears in each subset of the selected features.

### 2.5. Classification

A 5-fold stratified cross-validation was used for model validation, during which hyperparameters tuning was also performed. Bayesian optimization was selected for quicker hyperparameter tuning. The whole validation procedure was repeated ten times (inner loop) and the performance results were averaged, both on the inner loop and on the outer loop. Then, the optimized models were tested on the testing set and test performance metrics were averaged on the outer loop. Finally, six machine learning models, discriminant analysis (DA) [51], tree [52], K-nearest neighbors (KNN) [53], support vector machines (SVM) [54], Naïve Bayes (NB) [55], and ensemble [52] were trained, validated, and tested. For each model, the accuracy, AUC, sensitivity, specificity, precision, and the f-score were obtained.

#### Software Used for The Radiomics Analysis

matRadiomics [18], an IBSI compliant freeware, was used to perform a visual inspection of the images and segmentation, as well as performing the image pre-processing, feature extraction, and feature harmonization. MATLAB R2022b Update 1 was used to perform batch analysis, feature selection, and classification.

## 3. Results

### 3.1. Batch Analysis and Feature Harmonization

First, we conducted a preliminary batch analysis on all the datasets (4-classes dataset, 3-classes dataset, and 2-classes dataset) performing PCA and tSNE, whose plots show the presence of clusters, as illustrated in Figure 6, Figure 7 and Figure 8. The Stanford and VA groups were well separated from the MAASTRO group, with only few outliers, while a visual separation between the Stanford and VA groups could not be observed. Therefore, for each dataset, we performed a Kruskal–Wallis test on both of the principal components’ data to assess the statistical significance (*p*-value threshold = 0.05, null hypothesis, data in each group comes from the same distribution). Since the Kruskal–Wallis test suggested the statistical significance (*p* < 0.05)—both on the PC1 and PC2 data for all datasets, as shown in the Appendix A—we performed a post-hoc Tukey-HSD test to identify which groups were significantly different. For all the datasets, the post-hoc test confirmed that both the mean ranks of the Stanford and VA groups were significantly different from the mean rank of the MAASTRO group, both for the PC1 and PC2 data, while no significant difference was present between the mean ranks of the Stanford and VA groups, both on the PC1 (4-classes: *p* = 0.574, 3-classes: *p* = 0.6495, 2-classes: 0.5048) and PC2 (4-classes: *p* = 0.783, 3-classes: *p* = 0.5010, 2-classes *p* = 0.2289) data. Based on the batch analysis, we constructed the batch vector that was to be used for the feature harmonization. Since the only significant difference was found between the Stanford and MAASTRO groups, and also between the VA and MAASTRO groups we built our batch vector with only two classes, grouping Stanford and VA in a single class (1: MAASTRO and 2: Stanford + VA). Therefore, we performed the feature harmonization and verified if the procedure removed the batch effects (see Figure 9). The Kruskal–Wallis test suggested no significant difference both in the PC1 (4-classes: *p* = 0.137, 3-classes: *p* = 0.1193, 2-classes: *p* = 0.1665) and PC2 (4-classes: *p* = 0.964, 3-classes: *p* = 0.784, 2-classes: *p* = 0.3964) data, as is shown in the Appendix A.

### 3.2. Feature Selection and Stability

The feature selection process produced feature subsets of the different sizes at each outer loop iteration. Therefore, we reported, for each dataset, the minimum and the maximum size encountered, as shown in Table 4. Moreover, for each dataset, we reported the 20 features with the highest frequency, as is shown in the Appendix A. For all the datasets, the selected features had a *p*-value < 0.005. Moreover, the features that had a frequency greater than or equal to 80% belonged in a majority in relation to the texture class. Therefore, 11 features and 9 features in the 4-classes were in the non-harmonized and 4-classes harmonized datasets, respectively. In addition, 4 features were in both of the 3-classes’ non-harmonized and harmonized datasets, and 2 features were in both of the 2-classes harmonized and non-harmonized datasets, which were found to have a frequency greater than or equal to 80%. The details are shown in Table 5.

### 3.3. Classification

We reported the accuracy and the AUC that were averaged on the outer loop of the machine learning modeling pipeline for all the six different classifiers (DA, KNN, SVM, Naïve Bayes, Tree, and Ensemble), as well as for both of the non-harmonized and harmonized datasets in Figure 10, Figure 11, Figure 12 and Figure 13. Moreover, we reported in the Appendix A, the accuracy, AUC, sensitivity, specificity, precision, and f-score, together with the 95% confidence interval; in addition, the outer loop of the machine learning modeling pipeline for the classifiers that obtained the highest test accuracy was also averaged. Following this, SVM obtained the highest test accuracy (0.6141 ± 0.0317) in the 4-classes dataset; ensemble obtained the highest test accuracy (0.5624 ± 0.0555) in the 4-classes harmonized dataset; KNN obtained the highest test accuracy (0.6027 ± 0.0347) in the 3-classes dataset; ensemble obtained the highest test accuracy in the 3-classes harmonized dataset (0.5295 ± 0.0555); KNN obtained the highest test accuracy in the 2-classes dataset (0.7725 ± 0.0437); and Naïve Bayes obtained the highest accuracy (0.5857 ± 0.0418) in the 2-classes non-harmonized dataset. Furthermore, in Figure 14 and Figure 15, we reported examples of the ROCs (receiver operating characteristics), which were computed for both the harmonized and non-harmonized datasets, and which were also only for the classifiers that obtained the highest test accuracy.

## 4. Discussion

To date, most studies focusing on the differentiation of NSCLC subtypes have used private datasets or publicly available non-multicenter datasets. Meanwhile, only a few studies have focused on combining the datasets from different centers [33,34], although they did not investigate the presence of batch effects and the impact of feature harmonization on the models’ performances. Moreover, many studies have focused on the differentiation of only two NSCLC subtypes (SCC and ADC), without building a multiclass classifier [30,31,32,33,34]. Furthermore, due to the lack of large datasets, certain studies evaluated the performance of the classifiers only on the validation cohort, thereby resulting in a high risk of overfitting and low generalization.

Therefore, considering aims (i) and (ii), which were illustrated in Section 1.2, we investigated the possibility of building a multiclass classifier that would be able to differentiate between the four NSCLC subtypes (NOS, SCC, LCC, and ADC), based on two publicly available multicenter datasets. We evaluated the performance of the models both on the validation and on the testing cohorts, the impact of feature harmonization on models’ performance, and the performance variability that arose from adopting our modeling pipeline.

Our hypothesis that multicenter data could be affected by batch effects was confirmed by the experimental results, and, although ComBat feature harmonization is effective in removing it, the models’ performance of the harmonized datasets were always lower than those of the non-harmonized datasets. Moreover, a batch effect was present even if the voxel spacing was resampled to be isotropic, meaning that several causes could lead to the presence of batch effects in the multicenter studies, such as the kernel reconstruction and in kVp, among others. Furthermore, the different image segmentation methods adopted in the two datasets—manual for the NSCLC-Radiomics dataset, and semiautomatic with manual refinement for the NSCLC-Radiogenomics dataset—could have contributed to the data clusterization that was shown in the PCA scatter plot. Therefore, the combination of the two datasets may artificially increase the performance of the non-harmonized datasets, because the models learnt to distinguish between the two different segmentation methods. In any case, our experimental results show that the models’ accuracy was low. It was not possible to build a reliable and accurate model that was able to distinguish between the four different NSCLC subtypes with our proposed modeling pipeline (highest accuracy = 61.41%), even though an AUC equal to 0.831 was reached (and which was averaged on the 4 classes). Therefore, we decided to simplify the classification problem, reducing the dataset to three, and using only two classes. The worst results were obtained in the 3-classes’ datasets, both harmonized and non-harmonized (four classifiers out of six always showed a lower accuracy than those that were built using the 4-classes dataset), while the highest performance was obtained in the 2-classes datasets. In fact, the 2-classes non-harmonized dataset obtained the highest performance (highest accuracy = 77.25%) in comparison with the other datasets. However, similar to that which is mentioned above, this could be due to an artificial increase in performance, which is a result that is removed after feature harmonization (highest accuracy = 0.5857). Moreover, it is evident that the NOS class can be better separated (AUC ~ 0.96%) from the other classes (ADC, LCC, and SCC), and that this could be since ADC, LCC, and SCC have more physical properties in common. Meanwhile, the NOS class groups together all the types that do not have characteristics that are associated with the 3 main classes. The 3-classes’ datasets obtained the lowest accuracy because the more separable NOS class was removed.

Furthermore, it is not possible to assess which was the best classifier in absolute terms since this changed depending on the dataset being considered (i.e., 4-classes, 3-classes, 2-classes datasets) in both non-harmonized and harmonized data, as shown in the Section 3.

The third and fourth aims (see Section 1.2) of the proposed study was to evaluate the stability of the feature selection procedure. It is evident that the feature stability decreased as the number of classes decreased, and that the more stable features (i.e., a frequency ≥ 80%) were those that belonged to the wavelet and LoG image types and texture class, as is shown in several studies [56,57]. Furthermore, the selected features changed by varying the number of classes and, consequently, the amount of data, even when considering them between the harmonized and non-harmonized datasets. Therefore, it is difficult to establish in absolute terms which features contributed the most to the final models. In general, those with a frequency greater than or equal to 80% were more likely to have contributed the most.

The last aim of this study was to promote the need for reproducible and repeatable radiomics studies. Therefore, in Section 4.1, we discuss our results and the adopted modeling pipeline by comparing them with related works, while also considering the issues regarding the reporting of feature extraction parameters that were already discussed in Section 1.1 and Section 1.2.

### 4.1. Comparison of the Highest Accuracies and AUCs with Related Works

Table 6 and Table 7 summarize the existing works related to NSCLC classification problems in two groups: those whose aim is a multiclass classification (i.e., the four classes: NOS, ADC, LCC, and SCC) and those whose aim is a binary classification (i.e., the two classes: SCC and ADC). Table 6 shows that all the modeling methods (feature selection + classification) reached a test accuracy of ≥ 0.74, except for [31], which reached an accuracy that was equal to 0.656. The highest test accuracy and AUC (0.794, 0.863) were reached in study [32], in which a combination of ℓ2,1 norm regularization and linear discriminant analysis was used, while our method (Kruskal–Wallis + LASSO + KNN) obtained the second highest test accuracy and AUC (0.7725, 0.821). The differences between our work and [32] could be found in the machine learning methods adopted, in the dataset that was used (private datasets in [32] and multicenter datasets in our study), and in the images used. This is because they used PET/CT fused images, and due to the validation and testing schemes that were adopted. Indeed, we repeated both the training/testing splitting and the cross-validation procedure to obtain the average metrics that consider training and testing data variability. Moreover, due to having used multicenter data, a more robust performance was guaranteed. Indeed, a robust performance was obtained in the biggest multicenter study [33], in which the third highest accuracy and AUC (0.766, 0.815) were reached. Additionally, in that case, different classifiers were used and the synthetic minority over-sampling technique (SMOTE) was used to balance the training classes, thus leading to data expansion.

Other than the above, Table 7 shows that the best accuracy was reached in [36] (accuracy: 0.865, AUC: 0.747), but only on the validation set since a testing set was not available. Instead, our accuracy is closer to the accuracy reported in [35] when SMOTE was not used (overall testing accuracy ~ 0.67). Our proposed model obtained the lowest accuracy but the highest AUC (0.831) averaged on the 4 classes, and it was also the only multicenter study among them (see Section 1.1) Furthermore, in [35], the training set/testing set split (80/20%) was performed only once, and this could have led to more optimistic training and testing sets, especially when dealing with a not very large dataset. In addition, the SMOTE was used not only to balance the dataset but to also increase the number of samples. Nevertheless, in a real life clinical scenario, data are not balanced, and it is also crucial to test the model on an imbalanced dataset. Moreover, oversampling techniques, such as SMOTE, have certain drawbacks, with overfitting being the most common one [58]. However, the modeling pipeline proposed in our study was different, and the 10-repeated approach (inner/outer loops) was allowed to cope with the overoptimistic performance, which might be due to a good validation/test split, and provided information on how dataset composition influences selected features subsets, especially when small datasets are used (a common situation in the medical field). Moreover, results could be influenced by pre-processing parameters that strongly influence the feature extraction and model performance [59,60] and, as is shown in Section 1.1, they were not fully reported in the studies of [35,36]. Even if we used the same parameters of study [36] for the isotropic resampling and discretization, differences could be present in the resampling interpolator and in the wavelet method since they were not reported in the study itself.

As can be deduced, reproducibility is one of the major issues in radiomics studies. This is due to the different software, different ML methods, different choice of pre-processing parameters, and the different datasets being used. Moreover, if pre-processing parameters are not fully reported a complete comparison between radiomics studies cannot be carried out.

However, it seems that the accuracies were closer in the binary classification problems among all the studies, rather than in the multiclass classification problem. This could be due to the increased difficulty in differentiating between the four classes.

## 5. Limitation

The first limitation of this study was the dataset dimension. Indeed, even if we adopted all the strategies to reduce overfitting, the larger multicenter datasets are needed, especially for multiclass classification. Moreover, our 4-classes dataset was not balanced, and we did not adopt a strategy, such as SMOTE, to balance it. Furthermore, we showed the presence of the batch effects, but we did not extensively investigate their main cause. This could be due to different kernel reconstruction algorithms and kVp values. We also limited our study to only machine leaning techniques while deep learning is becoming much more popular and is also showing promising results in classifying NSCLC subtypes. Indeed, in study [32], it was shown how a VGG-16 convolutional neural network outperformed (accuracy: 0.841, AUC: 0.903) classical machine learning methods in differentiating between the wo NSCLC subtypes. Furthermore, in study [61], a self-supervised learning approach reached an AUC that was equal to 0.8641.

## 6. Conclusions

In our study, we investigated the possibility of building a multiclass radiomics model based on multicenter data that was capable of differentiating between the four different NSCLC subtypes. Unfortunately, when adopting our modeling pipeline a low accuracy was obtained (0.6141), thus meaning that an accurate mode—which is capable of differentiating between NOS, SCC, ADC, and LCC—could not be built when adopting our modelling pipeline, even if a high average AUC was obtained (0.831) when KNN is used as the classifier. A high accuracy (0.7725) and high AUC (0.821) were achieved when only two classes were considered and when the KNN method was used as the final classifier. The results also showed how it was difficult to identify which was the best classifier capable of differentiating between the NSCLC subtypes since it changed depending on the dataset used (i.e., the 4-classes, 3-classes, and 2-classes) for both harmonized and non-harmonized data. Moreover, to the best of our knowledge, this is the first study that has investigated the presence of batch effects and the impact of feature harmonization on multicenter radiomics CT-based models for the phenotyping of the four different NSCLC subtypes. Therefore, we showed that feature harmonization through ComBat was useful in terms of removing batch effects; however, the performance was found to always decrease, which is an aspect that should be further investigated in future studies. We also showed how isotropic resampling is not enough to homogenize images that come from different centers. Furthermore, as shown in Section 1.1, the comparison between radiomics studies is even more difficult since not all the parameters are fully reported in every study. The results are strongly influenced by the imaging pre-processing techniques, feature selection, and machine learning algorithms used during the radiomics analysis. Indeed, we showed that a modeling pipeline that works well for a task (such as binary classification) may not work well for another one (such as multiclass classification), thereby limiting the potentiality of radiomics. Moreover, it is important that all the parameters used in the pre-processing techniques are well reported because they have a strong impact on the extracted features. Additionally, dataset composition influences the performance results and selected features. Indeed, through our repeated modeling pipeline we showed that is difficult to find an absolute subset of selected features since it depends on the training/testing splitting. Therefore, we strongly believe that larger multicenter datasets are needed to build reliable models, as well as to reduce overfitting and to improve generalization. As for future directions, a deep investigation of why feature harmonization reduced the models’ performance should be investigated, together with the investigation of the main cause that generated the batch effects. Moreover, SMOTE should be introduced and tried in our modeling pipeline for data balancing, with the expectation of an increase in accuracy results. Finally, a deep learning approach using transfer learning and/or the newest self-supervised approaches should be pursued and compared to classical machine learning methods.

## Figures and Tables

**Figure 1 diagnostics-13-01167-f001:**
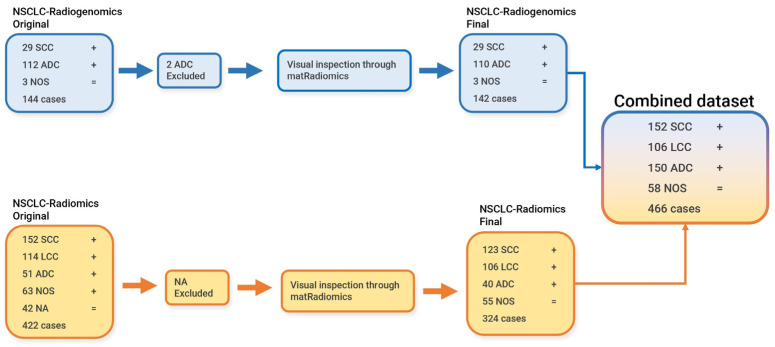
Data preparation workflow.

**Figure 2 diagnostics-13-01167-f002:**
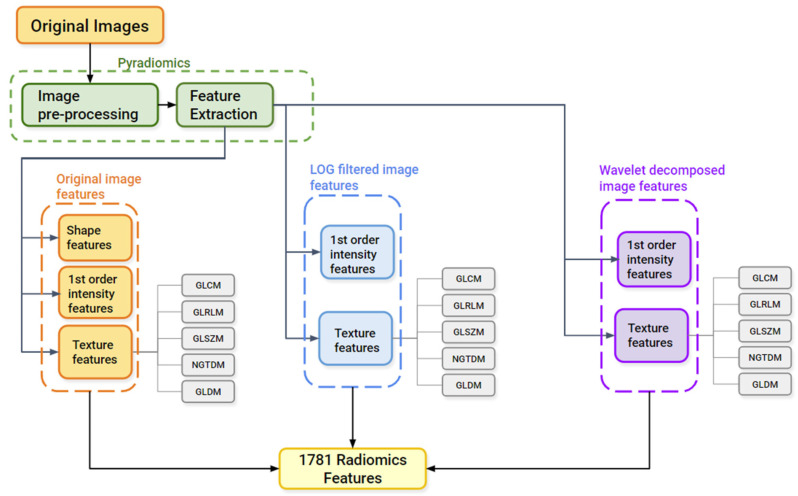
Feature extraction workflow.

**Figure 3 diagnostics-13-01167-f003:**
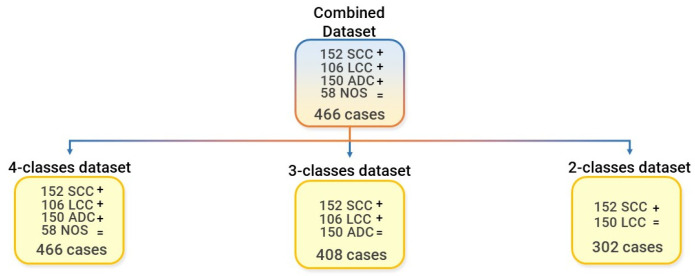
Final dataset subdivision into the 4-classes dataset, 3-classes dataset, and the 2-classes dataset.

**Figure 4 diagnostics-13-01167-f004:**
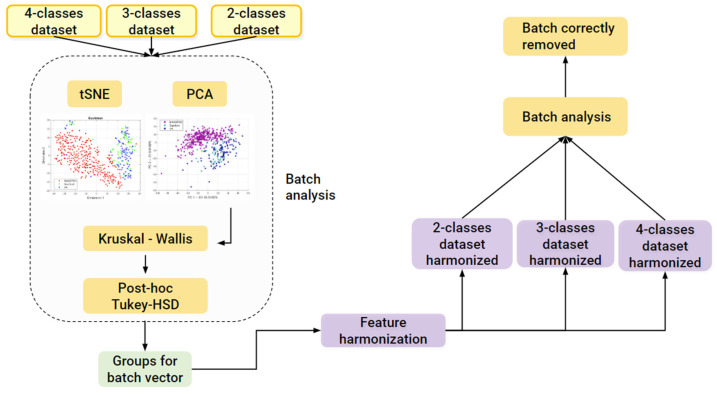
Batch analysis and harmonization pipeline.

**Figure 5 diagnostics-13-01167-f005:**
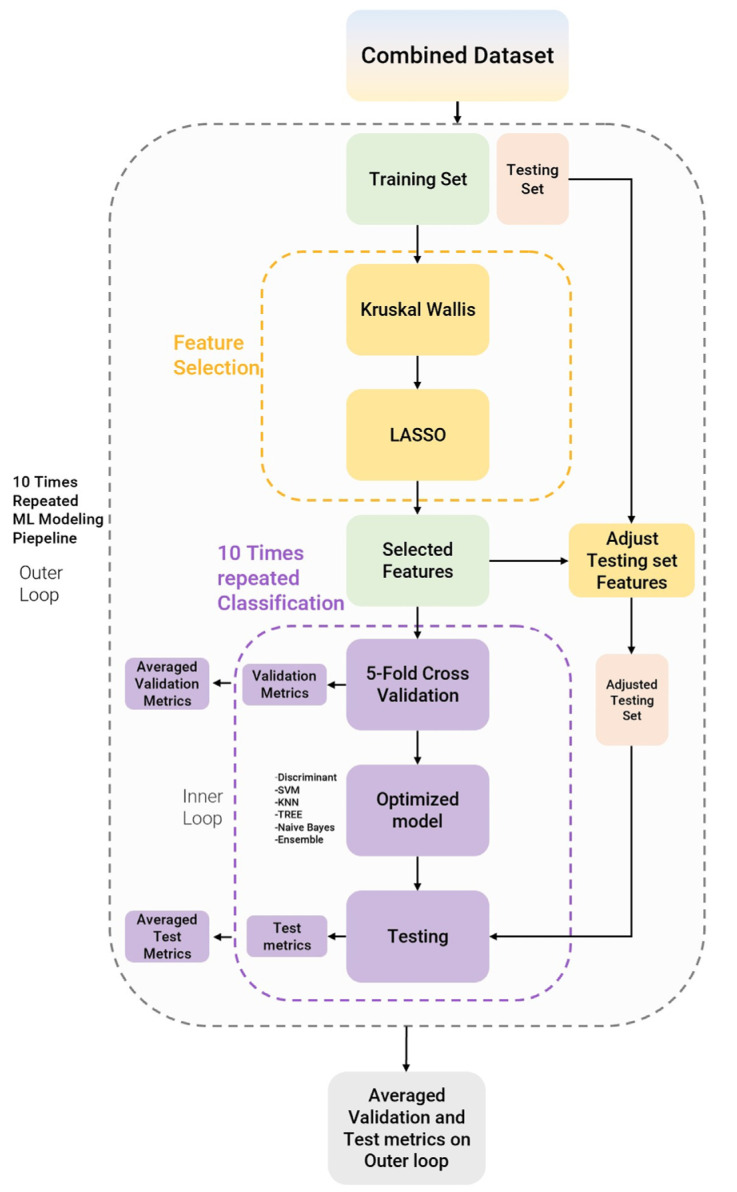
Modeling pipeline. The feature selection process is in yellow and the model building process is in purple.

**Figure 6 diagnostics-13-01167-f006:**
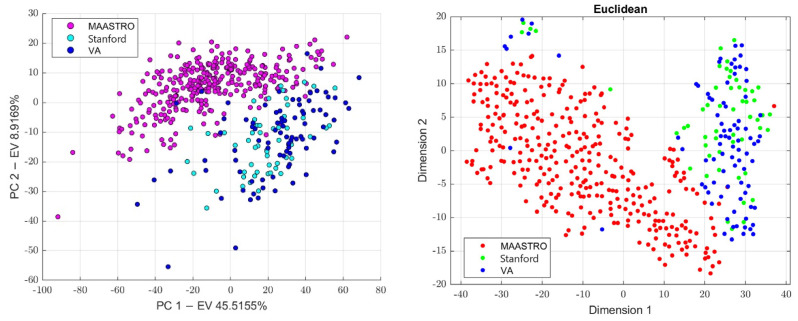
The PCA scatter plot on the (**left**) and the tSNE scatter plot on the (**right**) for the 4-classes dataset.

**Figure 7 diagnostics-13-01167-f007:**
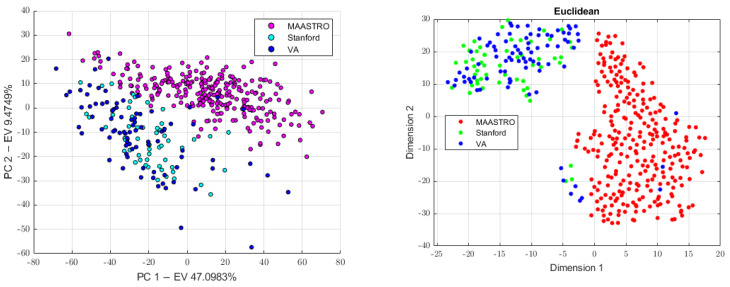
The PCA scatter plot on the (**left**) and the tSNE scatter plot on the (**right**) for the 3-classes dataset.

**Figure 8 diagnostics-13-01167-f008:**
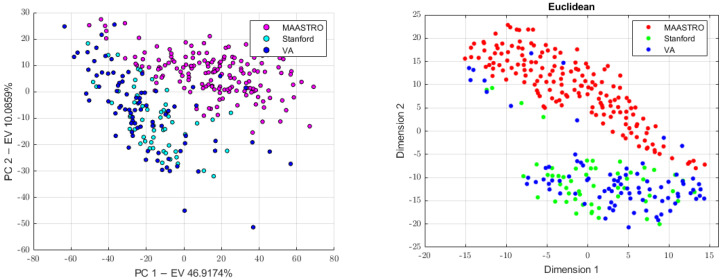
The PCA scatter plot on the (**left**) and the tSNE scatter plot on the (**right**) for the 2-classes dataset.

**Figure 9 diagnostics-13-01167-f009:**
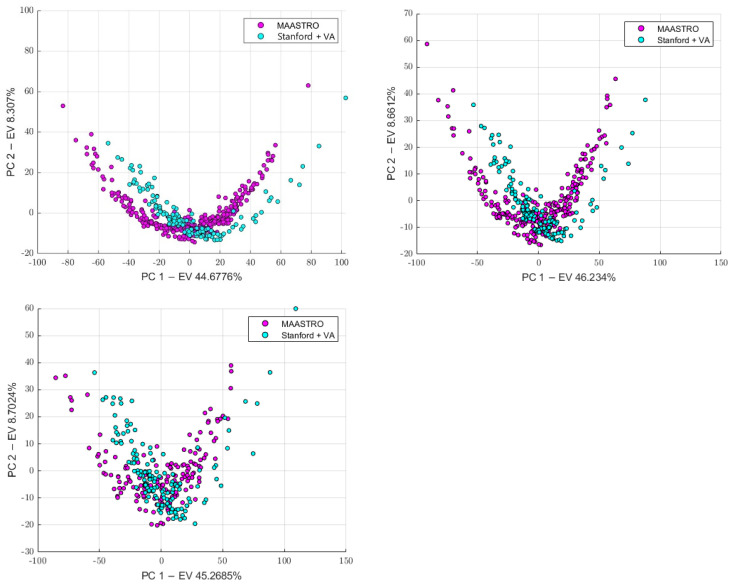
The PCA scatter plot after harmonization. The 4-classes are on the **top-left**, the 3-classes are on the **top-right**, the 2-classes are on the **bottom-left**.

**Figure 10 diagnostics-13-01167-f010:**
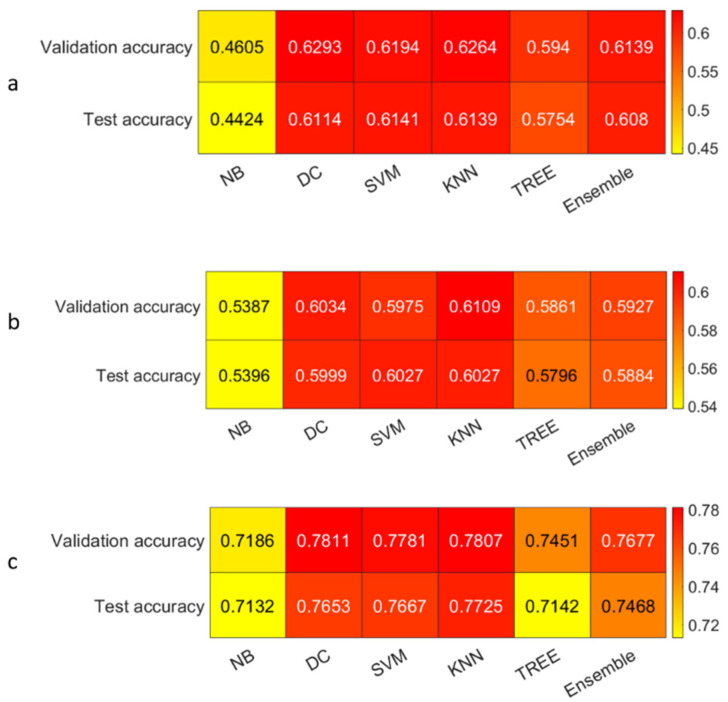
Accuracy heatmaps: (**a**) The 4-classes datasets, (**b**) 3-classes datasets, and (**c**) 2-classes datasets.

**Figure 11 diagnostics-13-01167-f011:**
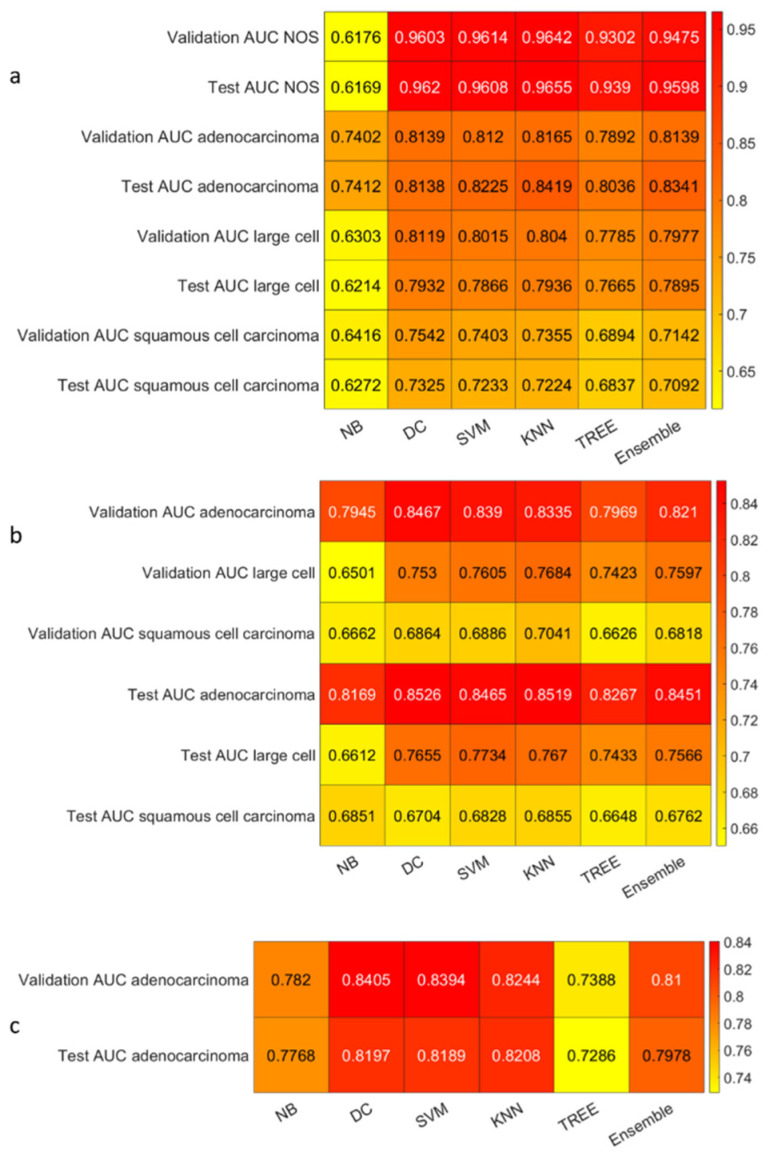
AUC heatmaps: (**a**) The 4-classes datasets, (**b**) 3-classes datasets, and (**c**) 2-classes datasets.

**Figure 12 diagnostics-13-01167-f012:**
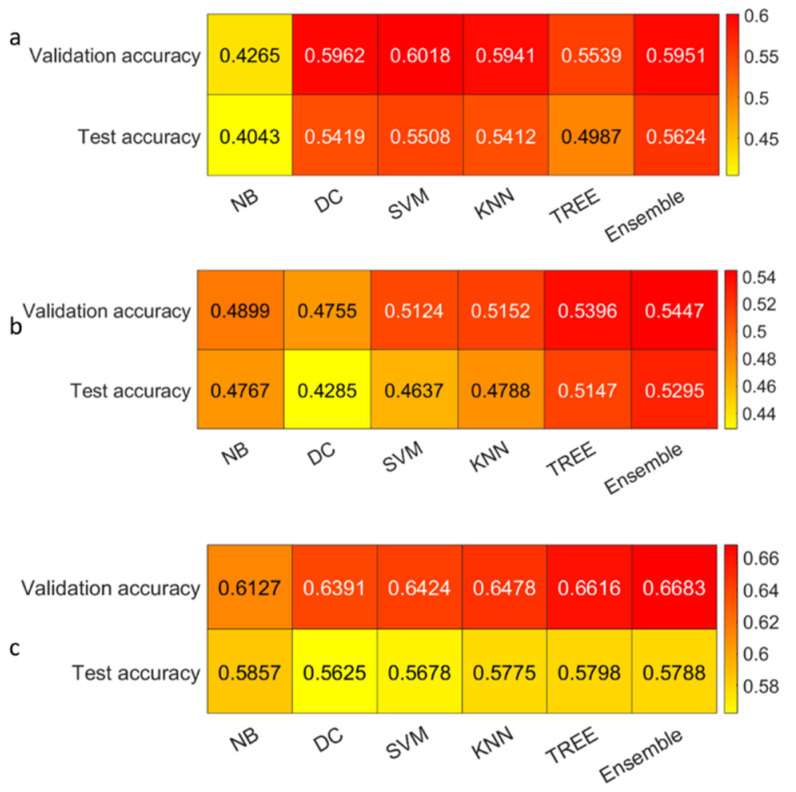
Accuracy heatmaps for the harmonized datasets: (**a**) The 4-classes datasets, (**b**) 3-classes datasets, and (**c**) 2-classes datasets.

**Figure 13 diagnostics-13-01167-f013:**
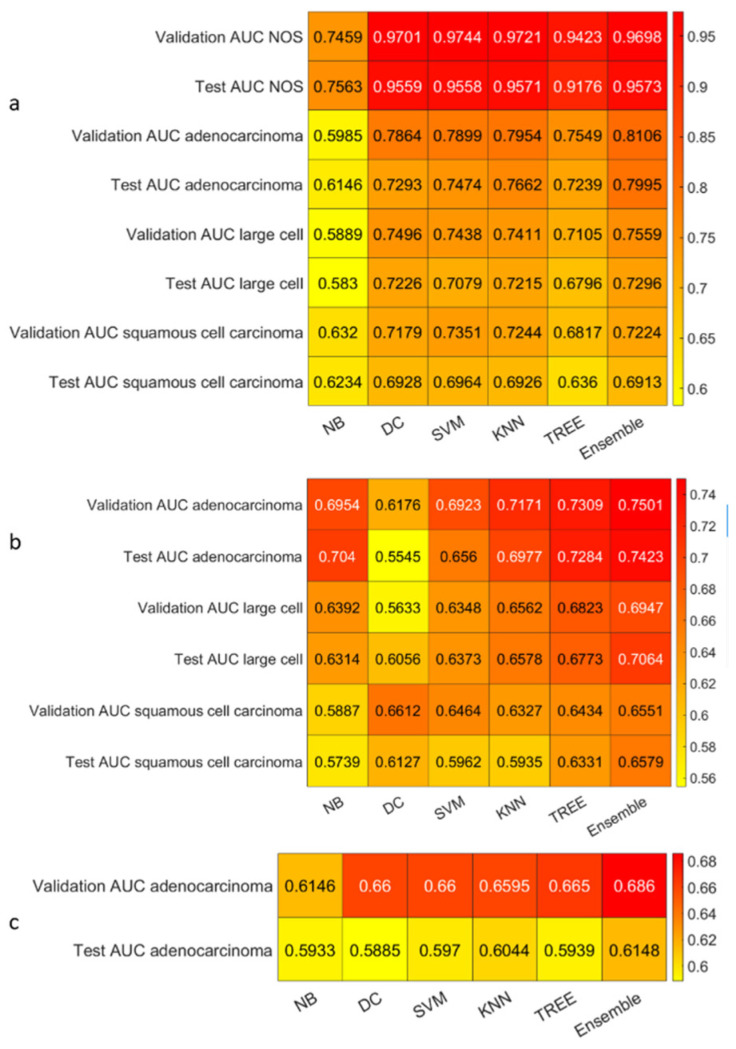
AUC heatmaps for the harmonized datasets: (**a**) The 4-classes datasets, (**b**) 3-classes datasets, and (**c**) 2-classes datasets.

**Figure 14 diagnostics-13-01167-f014:**
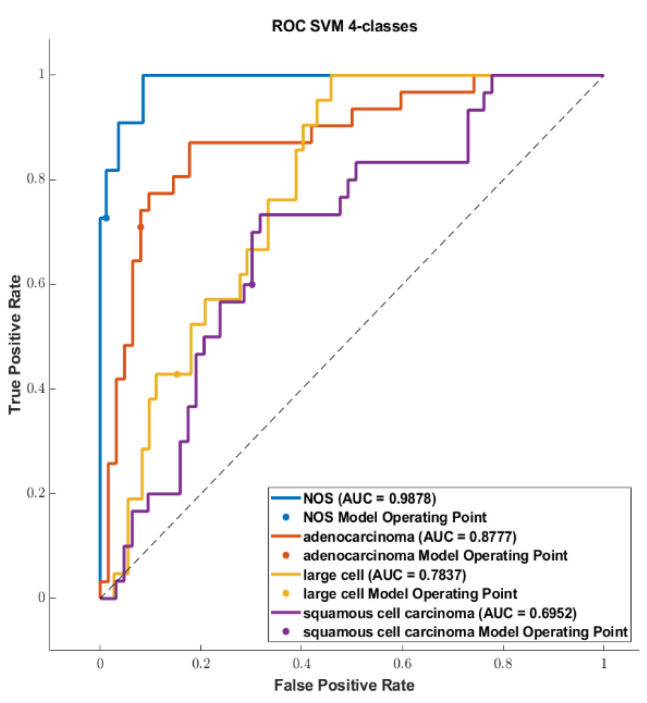
ROC curves: (**top**) The 4-classes datasets, (**middle**) 3-classes datasets, and (**bottom**) 2-classes datasets.

**Figure 15 diagnostics-13-01167-f015:**
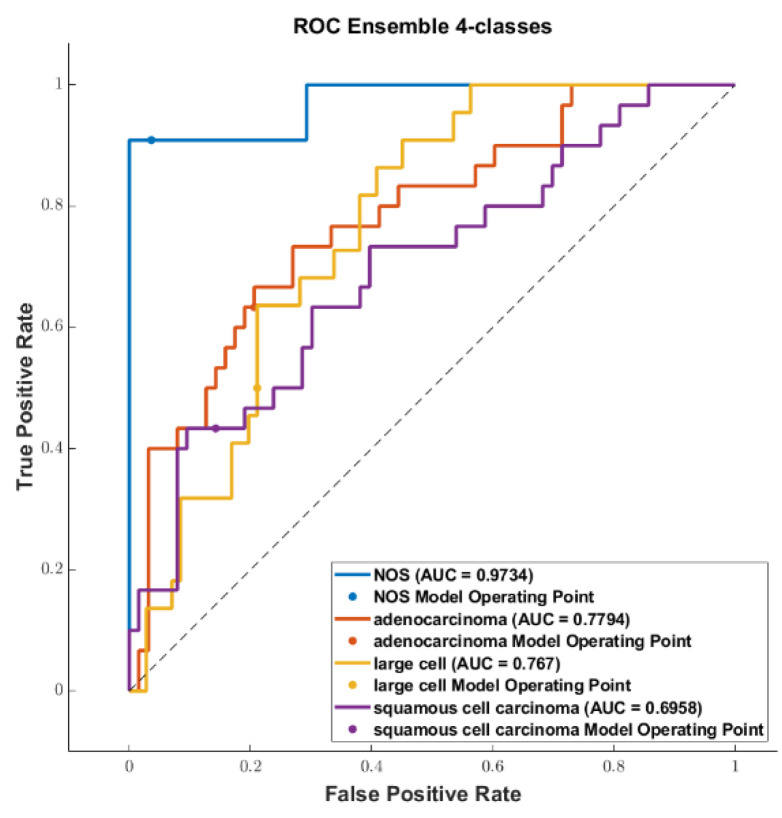
ROC curves for the harmonized datasets: (**top**) The 4-classes datasets, (**middle**) 3-classes datasets, and (**bottom**) 2-classes datasets.

**Table 1 diagnostics-13-01167-t001:** The binary classification studies (SCC vs. ADC). M: multicenter, B: batch analysis, and H: harmonization. The number in the round brackets in the dataset column refers to the number of patients that were involved after a selection from the total available amount.

Work	Dataset	Modality	Feature Extraction	M/B/H
[30] (2016)	1 public dataset [38] (192) and lung 2 (152)	CT	Pre-processing: not all specified Extractor: Matlab 2012 Extracted Features: shape, 1st-order statistics, texture features (440)	No/no/no
[31] (2017)	Private dataset (40)	CT	Pre-processing: all specified Extractor: Mvalliers package Extracted Features: shape, 1st-order statistics, texture features (476)	No/no/no
[32] (2021)	Private dataset (1419)	PET + CT	Pre-processing: not all specified Extractor: PyRadiomics Extracted Features: 1st-order statistics, texture Features (688)	No/no/no
[33] (2021)	Private dataset (302) and 2 public datasets [38] (203), [39] (140)	CT	Pre-processing: not all specified Extractor: PyRadiomics Extracted Features: shape, 1st-order statistics, texture features (788)	Yes/no/no
[34] (2023)	8 public datasets [38,39,40,41,42,43,44,45] (868).	CT	Pre- processing: not all specified, interpolator specified Extractor: PyRadiomics Extracted Features: shape, 1st-order statistics, texture features (1409).	Yes/no/ no
ours	2 public datasets: [38,39] (302)	CT	Pre-processing: all specified Extractor: PyRadiomics Extracted Features: shape, 1st-order statistics, texture features (1433)	Yes/Yes/Yes

**Table 2 diagnostics-13-01167-t002:** Multiclass classification studies. M: multicenter, B: batch analysis, and H: harmonization. The number in the round brackets in the dataset column refers to the number of patients that were involved after a selection from the total available amount.

Work	Dataset	Modality	Feature Extraction	M/B/H
[35] (2019)	2 public datasets [38] (278), [45] (71)	CT	Pre-processing: not all specified. Extractor: not specified Extracted Features: shape, 1st-order statistics, texture features (440)	No/no/no
[36] (2021)	1 public dataset: [38] (354)	CT	Pre-processing: not all specified, Extractor: PyRadiomics Extracted Features: shape, 1st-order statistics, texture features (1433)	No/no/no
ours	2 public datasets: [38,39] (466)	CT	Pre-processing: all specified Extractor: PyRadiomics Extracted Features: shape, 1st-order statistics, texture features (1433)	Yes/Yes/Yes

**Table 3 diagnostics-13-01167-t003:** The pixel spacing (mm), slice thickness (mm), and matrix dimension of each CT scan.

Datasets	Pixel Spacing [x, y] (n°)	Slice Thickness [z] (n°)	Matrix Dimension [x, y] (n)°
NSCLC-Radiomics	[0.97656250, 9765625] (247)	3 (247)	[512 × 512] (247)
NSCLC-Radiomics	[0.9770, 0.9770] (77)	3 (77)	[512 × 512] (77)
NSCLC-Radiogenomics	In a range between [0.589844, 0.976562] (142)	In a range between [0.625, 3] (142)	[512 × 512] (142)

**Table 4 diagnostics-13-01167-t004:** Minimum and maximum size subsets for each dataset.

Size	4-c-nh	4-c-h	3-c-nh	3-c-h	2-c-nh	2-c-h
Minimum	16	14	9	2	8	1
Maximum	28	22	23	14	22	9

**Table 5 diagnostics-13-01167-t005:** Selected features with a frequency greater than or equal to 80%.

		4-c-nh	4-c-h	3-c-nh	3-c-h	2-c-nh	2-c-h
	Amount	11	9	4	2	2	1
	Shape	3	1	0	0	0	0
Class	First order	1	1	1	0	0	0
	Texture	**7**	**7**	**3**	**2**	**2**	**1**
	Original	3	1	0	0	0	0
Image Type	LoG	**4**	1	0	0	0	**1**
	Wavelet	**4**	**7**	**4**	**2**	**2**	**0**

The highest values of interest are highlighted in bold.

**Table 6 diagnostics-13-01167-t006:** A binary classification (SCC and ADC) comparison between the works, in which machine learning methods, validation and testing schemes, as well as the best averaged results are reported. Acronyms: Gini index (GINI), information gain (IG), gain ratio (GR), minimum description length (MDL), DKM (author names), Laplacian score (LS), spectral feature selection (SPEC), ℓ2,1-norm regularization (ℓ2,1NR), efficient and robust feature selection (RFS), multi-cluster feature selection (MCFS), chi-square score (CSS), Fisher score based on statistics (FS), t-score (TS), redundancy maximum relevance feature selection (mRMR), sequential forward selection (SFS), and least absolute shrinkage and selection operator (LASSO), random forest (RF), Naïve Bayes (NB), Gaussian Naïve Bayes (GNB), K-nearest neighbors (KNN), AdaBoost (AdaB), extreme gradient boosting (XGBoost), bagging (BAG), decision tree (DT), gradient boosting decision tree (GDBT), logistic regression (LR), multilayer perceptron (MLP), linear discriminant analysis (LDA), and support vector machines (SVM).

Work	ML Methods	Training and Testing Sets	Validation and Testing Schemes	Results
[30] (2016)	Selection: correlation + GINI, IG, GR, MDL, DKM, ReliefF + variants. Classification: RF, NB, and KNN	Training: 192 (public dataset) Testing: 152 (Lung 2)	External testing	ReliefFdistance + NB: Test AUC 0.72
[31] (2017)	Selection: univariate analysis *p* < 0.05 + interobserver variation analysis *p* < 0.1 + cross correlation analysis r < 0.7. Classification: NB	Training: 28 (private dataset) Testing: 12 (private dataset)	Training/testing sets splitting: 30 times repeated (70/30)	Test accuracy: 0.656 Test AUC: 0.725
[32] (2021)	Selection: LS, ReliefF, SPEC, ℓ2,1NR, RFS, MCFS, CSS, FS, TS, and GINI. Classification: AdaB, BAG, DT, NB, KNN, LR, MLP, LDA, and SVM	Training: 1136 (private dataset) Testing: 283 (private dataset)	Training/testing sets splitting: (80/20). Validation: 10-fold cross validation	ℓ2,1NR + LDA: test accuracy 0.794 ℓ2,1NR + LDA: test AUC 0.863
[33] (2021)	Selection: mRMR, SFS, and LASSO Classification: LR, SVM, and RF	Since the ratio of training/testing split is not specified (merged dataset), the number of samples used for training and testing is unknown	Training/testing sets splitting: ratio not specified Validation: 5-fold cross validation	Test accuracy 0.74 Test AUC 0.78
[34] (2023)	Selection: wrapper ℓ2,1 norm minimization + 10-fold cross validated LR Classification: LR, SVM, RF, MLP, KNN, GNB, GBDT, AdaB, BAG, and XGBoost	Training: from 560 (5 merged datasets) to 940 after SMOTE. Testing: 140 (5 merged datasets) + external testing (168-3 datasets)	Training/Testing sets splitting: (80/20) Validation: 10-fold cross validation External testing	Bagging-AdaBoost-LR (ensemble): test accuracy 0.766 Bagging-AdaBoost-SVM (Ensemble): test AUC 0.815
ours	Selection: Kruskal Wallis + LASSO Classification: NB, DA, KNN, SVM, TREE, and ensemble	Training: 242 (2 datasets merged) Testing: 60 (2 datasets merged)	Training/Testing sets splitting: 10 times repeated (80/20) Validation: 10 times repeated 5-fold cross validation	KNN: test accuracy 0.7725 KNN: test AUC 0.821

**Table 7 diagnostics-13-01167-t007:** Multiclass classification comparison between works, in which machine learning methods, validation and testing schemes, and the best averaged results are reported. Acronyms: least absolute shrinkage and selection operator (LASSO), Naïve Bayes (NB), K-nearest neighbors (KNN), discriminant analysis (DA), random forest (RF), and support vector machines (SVM).

Work	ML Methods	Training and Testing Sets	Validation and Testing Schemes	Results
[35] (2019)	Selection: wrapper ℓ2,1 norm minimization + SVM Classification: SVM	Training: from 279 (public dataset) to 760 after SMOTE Testing: 70 (public dataset)	Training/Testing sets splitting: (80/20) Validation: 10-fold cross validation	NO SMOTE test accuracy 0.67 SMOTE Test accuracy 0.86
[36] (2021)	Selection: Wrapper algorithm, multivariate adaptive regression splines Classification: Multinomial logistic regression	Training: 354 (public dataset)	Validation: 1000 bootstrapping	Validation accuracy: 0.865 Validation AUC: 0.747
ours	Selection: Kruskal Wallis + LASSO Classification: NB, DA, KNN, SVM, TREE, and ensemble	Training: 373 (2 datasets merged) Testing: 93 (2 datasets merged)	Training/Testing sets splitting: 10 times repeated (80/20) Validation: 10 times repeated 5-fold cross validation	DC: validation accuracy 0.6293 SVM: test accuracy 0.6141 DC: validation AUC 0.826 (averaged on 4 classes) KNN: test AUC 0.831 (averaged on 4 classes)

## Data Availability

Not applicable.

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
