# Peer review of "Phenotyping the Histopathological Subtypes of Non-Small-Cell Lung Carcinoma: How Beneficial Is Radiomics?"

_diagnostics, 2023, doi:10.3390/diagnostics13061167_

Round 1

Reviewer 1 Report

1. Precise the Abstract and Conclusion and novelty should be highlighted. 

2. All acronyms should be given in a Table in Appendix. An extensive literature reviews should be added why your work is need and comparisons must be given in a separate section. 

3. Motivation should be clear in Introduction. 

4. Section 2.3.2. Image pre-processing and feature extraction is not clear according to Fig. 2. 

5. Caption of Figure 5 should be improved.

6. According to Fig. 4, why 3-classes dataset harmonized and 2-classes dataset harmonized have no batch analysis?

7. Mathematical flow and proof of proposed method are not provided in the work. Clarity and presentation should be improved. 

8. Resolution of Figure 8, 9 should be improved. 

9. Compare the work with state-of-the-arts and existing literatures.

10. Author should cite the following references in this paper:

(i) Huang, Y., Jiang, X., Xu, H., Zhang, D., Liu, L.N., Xia, Y.X., Xu, D.K., Wu, H.J., Cheng, G. and Shi, Y.H., 2023. Preoperative prediction of mediastinal lymph node metastasis in non-small cell lung cancer based on 18F-FDG PET/CT radiomics. Clinical Radiology78(1), pp.8-17.

(ii) Song, F., Song, X., Feng, Y., Fan, G., Sun, Y., Zhang, P., Li, J., Liu, F. and Zhang, G., Radiomics feature analysis and model research for predicting histopathological subtypes of non‐small cell lung cancer on CT images: A multi‐dataset study. Medical Physics.

(iii) Bharati S, Podder P, Mondal MR. Hybrid deep learning for detecting lung diseases from X-ray images. Informatics in Medicine Unlocked. 2020 Jan 1;20:100391.

(iv) Dwivedi K, Rajpal A, Rajpal S, Agarwal M, Kumar V, Kumar N. An explainable AI-driven biomarker discovery framework for Non-Small Cell Lung Cancer classification. Computers in Biology and Medicine. 2023 Jan 12:106544.

11. Future directions should be added. 

12. Overall presentation should be improved with highlight the originality. 

Author Response

Dear reviewer, thank you for all your valuable suggestions and comments. Below, we provided a point-to-point answer. We considered your suggestions and made many additions to the manuscript, which we think was considerably improved. All the changes to the manuscript have been highlighted in yellow.

1.Precise the Abstract and Conclusion and novelty should be highlighted. 

Thank you for your valuable comment. We have modified the Abstract and Conclusion to emphasize the novelty of the proposed study. Specifically, the main novelty is an in-depth analysis of the performance results as the radiomics model varies for the phenotyping of four different NSCLC subtypes in a multi-center study. In addition, to better highlight research motivation and contribution, we have added the “Research motivation and contribution” Section 1.2 in the manuscript.

2. All acronyms should be given in a Table in Appendix. An extensive literature reviews should be added why your work is need and comparisons must be given in a separate section. 

Thank you for your valuable comment. We added Section 1.1 in which we made a preliminary comparison between the cited studies. We compared the type of the study (multicenter or not multicenter), the presence of batch effects analysis and feature harmonization. Moreover, we compared the reporting of feature extraction parameters (complete and not complete). Therefore, we added Table 1 and 2 in which we summarized the characteristics of the considered studies. Then, we completed the comparison between related works in section 5.1 in which we added tables 6 and 7, and we compared machine learning methods, validation schemes and results between related works. Moreover, we added all the acronyms in a table in Appendix B.

3. Motivation should be clear in Introduction. 

Thank you for this comment. We added Section 1.2 in which we reported research motivation and contribution of our study.

4. Section 2.3.2. Image pre-processing and feature extraction is not clear according to Fig. 2.

Thank you for this comment. We have remade the Fig 2.

5. Caption of Figure 5 should be improved.

Thank you for this comment. We have changed the caption of Figure 5.

6. According to Fig. 4, why 3-classes dataset harmonized and 2-classes dataset harmonized have no batch analysis?

Thank you for this question which made us realize that the batch analysis and feature harmonization sections were not clear enough. Therefore, we added the batch analysis for the 3-classes and 2-classes datasets, and consequently we modified Sections 2.3.4 and 3.1 rewriting some phrases to make both sections and batch analysis as clear as possible and we added three more Figures (7, 8 ,9). Moreover, we modified Figure 4 and the Supplementary Materials file. Including the batch analysis for the 3-classes and 2-classes datasets reinforces the choice of having used only 2 centers rather than 3 for the construction of the batch vector, as statistical analysis also suggest.

7. Mathematical flow and proof of proposed method are not provided in the work. Clarity and presentation should be improved. 

Dear Reviewer, we hope we've made our study more comprehensive and made the intent of our goal more explicit, thanks to your suggestions and the changes we've introduced. Specifically, from figure 1 to figure 5 we describe the whole flow of the proposed method.

8. Resolution of Figure 8, 9 should be improved. 

Thank you for this comment. Since the resolution of both Figure 8 and 9 was poor, we divided the AUC from the accuracy by adding Figures 10, 11, 12, and 13.

9. Compare the work with state-of-the-arts and existing literatures.

Thank you for this valuable comment. We added a preliminary comparison between studies in Section 1.1 in which we compared the type of the study (multicenter or not multicenter), the presence of batch effects analysis, feature harmonization and the reporting of feature extraction parameters. Therefore, we added Tables 1 and 2 in which we summarized the characteristics of the considered studies. Furthermore, we added Section 5.1 in which we compared machine learning methods, validation schemes, and results with other related works.

10. Author should cite the following references in this paper:

(i) Huang, Y., Jiang, X., Xu, H., Zhang, D., Liu, L.N., Xia, Y.X., Xu, D.K., Wu, H.J., Cheng, G. and Shi, Y.H., 2023. Preoperative prediction of mediastinal lymph node metastasis in non-small cell lung cancer based on 18F-FDG PET/CT radiomics. Clinical Radiology78(1), pp.8-17.

(ii) Song, F., Song, X., Feng, Y., Fan, G., Sun, Y., Zhang, P., Li, J., Liu, F. and Zhang, G., Radiomics feature analysis and model research for predicting histopathological subtypes of nonsmall cell lung cancer on CT images: A multidataset study. Medical Physics.

(iii) Bharati S, Podder P, Mondal MR. Hybrid deep learning for detecting lung diseases from X-ray images. Informatics in Medicine Unlocked. 2020 Jan 1;20:100391.

(iv) Dwivedi K, Rajpal A, Rajpal S, Agarwal M, Kumar V, Kumar N. An explainable AI-driven biomarker discovery framework for Non-Small Cell Lung Cancer classification. Computers in Biology and Medicine. 2023 Jan 12:106544.

Thank you for this valuable suggestion. We added all these references in the manuscript.

11. Future directions should be added. 

Thank you for this valuable suggestion. We added future directions in the Conclusion section.

12. Overall presentation should be improved with highlight the originality. 

Dear Reviewer, thank you for this valuable comment. We have rewritten the Abstract and added section 1.2 to report the motivations of the research, highlighting the originality of our work. Additionally, we have added section 1.1 to compare our study with related work and edited the figures to make them clearer. Finally, we have modified the Discussion and Conclusion sections to critically discuss the obtained results and to add future research directions. Therefore, we hope that we have improved the readability of the work thanks to your suggestions and the many changes we have introduced.

Reviewer 2 Report

This paper is about Phenotyping the histopathological subtypes of non-small cell lung carcinoma: which is the benefit of Radiomics?. The novelty of this paper is unclear for me. However, My comments is as follows:
1. The abstract section should be edited. In this section, the authors should provide important information such as funding, etc.
2. In introduction section, Please explain more about different deep learning methods. In the following, I recommended the https://doi.org/10.1016/j.inffus.2022.12.010, https://arxiv.org/abs/2105.04881 , https://doi.org/10.1007/978-3-031-06242-1_7references for this section. In these references, all deep learning methods are presented and help you.
3. The literature review is not comprehensive. It is better to have a tabular summary of the paper's review to give readers a better understanding of the research done in this field. In this section, some articles should be presented in the form of text, and the rest of the articles should be summarized in the table with this information (Works, Dataset, preprocessing, AI methods, validation, and Performance (%)). Please review more than 25 new papers (2021-2023).
4. Please show the ROC and loss curves for your model.
5. The research question(s) need to appear stronger and clearer.
6. Please clarify your initial hypothesis.
7. In discussions you need to critically discuss your work/results against your hypothesis.
8. Identify the main findings and justify the novelty and contribution of the work.
9. A recap of all the relevant parameters with their meaning should be added to help the reader..

10. Please add a section about "limitation of study".
11. In the Conclusion section, please explain more about future works. This section requires further discussion.. For example, you can discuss about new deep learning methods such as transformers, attention learning, and etc.
12. Please add a table in conclusion and compare your proposed method with another related works.
13. English language is acceptable in general, but there are some errors that should be corrected.

Author Response

Dear reviewer, thank you for all your valuable suggestions and comments. Below, we provided a point-to-point answer. We considered your suggestions and made many additions to the manuscript, which we think was considerably improved. All the changes to the manuscript have been highlighted in yellow.

This paper is about Phenotyping the histopathological subtypes of non-small cell lung carcinoma: which is the benefit of Radiomics?. The novelty of this paper is unclear for me. However, my comments is as follows:

1.The abstract section should be edited. In this section, the authors should provide important information such as funding, etc.

Thank you for your valuable comment. We have modified the Abstract and added the main findings. Specifically, we have emphasized the novelty of the proposed study, namely an in-depth analysis of the performance results as the radiomics model varies for the phenotyping of four different NSCLC subtypes in a multi-center study. In addition, to better highlight research motivation and contribution, we have added the “Research motivation and contribution” Section 1.2 in the manuscript.

2. In introduction section, Please explain more about different deep learning methods. In the following, I recommended the https://doi.org/10.1016/j.inffus.2022.12.010, https://arxiv.org/abs/2105.04881 , https://doi.org/10.1007/978-3-031-06242-1_7references for this section. In these references, all deep learning methods are presented and help you.

Thank you for this valuable suggestion. Since we wanted to keep our work related to radiomics as much as possible, we added references https://doi.org/10.1016/j.inffus.2022.12.010 and https://arxiv.org/abs/2105.04881 to our manuscript.

3. The literature review is not comprehensive. It is better to have a tabular summary of the paper's review to give readers a better understanding of the research done in this field. In this section, some articles should be presented in the form of text, and the rest of the articles should be summarized in the table with this information (Works, Dataset, preprocessing, AI methods, validation, and Performance (%)). Please review more than 25 new papers (2021-2023).

Thank you for this comment. The aim of this paper is not a review of the current radiomics state of art, and we think that finding and comparing 25 papers related to NSCLC subtypes classification, radiomics and classical machine learning is out of the purpose of our work. However, we understood that the cited literature related to our work was not comprehensive and that a clearer comparison between related works were needed. Therefore, we added some references related to our work, and we added Section 1.1 in which we made a preliminary comparison between the cited studies. We compared the type of the study (multicenter or not multicenter), the presence of batch effects analysis and feature harmonization. Moreover, we compared the reporting of feature extraction parameters (complete or not complete). Therefore, we added Table 1 and 2 in which we summarized these characteristics. Then, we completed the comparison between related works in section 5.1 in which we added tables 6 and 7, and we compared machine learning methods, validation schemes and results between related works.

4. Please show the ROC and loss curves for your model.

Thank you for this comment. We have added Figures 14 and 15.

5. The research question(s) need to appear stronger and clearer.

Thank you for this comment. We have added the “Research motivation and contribution” Section to report research motivation and contribution of our study.

6. Please clarify your initial hypothesis.

Thank you for your valuable comment. We have modified the Abstract to emphasize our initial hypothesis, i.e. the effective utility of radiomics as the radiomics models vary.

7. In discussions you need to critically discuss your work/results against your hypothesis.

Thank you for this valuable comment. Considering Section 1.2, we have rewritten the Discussion section. We have also added Section 5.1, in which the comparison between our work and related works was discussed.

8. Identify the main findings and justify the novelty and contribution of the work.

Thank you for your valuable comment. The main novelty is an in-depth analysis of the performance results as the radiomics model varies for the phenotyping of four different NSCLC. Furthermore, we considered the batch analysis and the impact of feature harmonization subtypes in a multi-center study. To better highlight research motivation and contribution, we added Section 1.1 and 1.2 in the manuscript. Moreover, we rewrote the Conclusion section in which we reported the main findings.

9. A recap of all the relevant parameters with their meaning should be added to help the reader..

Thank you for this valuable comment. We added a recap of all the relevant parameters for feature extraction in Table 1A in Appendix A.

10. Please add a section about "limitation of study".

Thank you for this valuable comment. We added Section 6 called Limitation.

11. In the Conclusion section, please explain more about future works. This section requires further discussion.. For example, you can discuss about new deep learning methods such as transformers, attention learning, and etc.

Thank you for this valuable comment. We rewrote the Conclusion section, in which we reported the main findings and contribution of our work and added future directions.

12. Please add a table in conclusion and compare your proposed method with another related works.

Thank you for this valuable comment. We compared our work with related works in section 5.1.

13. English language is acceptable in general, but there are some errors that should be corrected.

Thank you for this comment. We revised the manuscript and corrected as many English language errors as we could find.

Round 2

Reviewer 1 Report

The current version is ready for the publication. 

Reviewer 2 Report

Thanks for your revision. I strongly recommend that this version of the paper can be accepted.